# Lidar–Camera Semi-Supervised Learning for Semantic Segmentation

**DOI:** 10.3390/s21144813

**Published:** 2021-07-14

**Authors:** Luca Caltagirone, Mauro Bellone, Lennart Svensson, Mattias Wahde, Raivo Sell

**Affiliations:** 1Applied Artificial Intelligence Research Group, Department of Mechanics and Maritime Sciences, Chalmers University of Technology, 412 58 Gothenburg, Sweden; luca.caltagirone@protonmail.com (L.C.); mattias.wahde@chalmers.se (M.W.); 2Smart City Center of Excellence, Tallinn University of Technology, 12616 Tallinn, Estonia; 3Department of Electrical Engineering, Chalmers University of Technology, 412 58 Gothenburg, Sweden; lennart.svensson@chalmers.se; 4Department of Mechanical and Industrial Engineering, Tallinn University of Technology, 12616 Tallinn, Estonia; raivo.sell@taltech.ee

**Keywords:** sensor fusion, semi-supervised learning, deep learning, semantic segmentation

## Abstract

In this work, we investigated two issues: (1) How the fusion of lidar and camera data can improve semantic segmentation performance compared with the individual sensor modalities in a supervised learning context; and (2) How fusion can also be leveraged for semi-supervised learning in order to further improve performance and to adapt to new domains without requiring any additional labelled data. A comparative study was carried out by providing an experimental evaluation on networks trained in different setups using various scenarios from sunny days to rainy night scenes. The networks were tested for challenging, and less common, scenarios where cameras or lidars individually would not provide a reliable prediction. Our results suggest that semi-supervised learning and fusion techniques increase the overall performance of the network in challenging scenarios using less data annotations.

## 1. Introduction

Nowadays, data are considered a valuable asset generating massive investments. However, how much data should autonomous vehicles collect to generate a reasonable driving model? Currently, Waymo has a fleet composed of around 55 vehicles driving over 1 million kilometres per year, roughly corresponding to 30,000 h. This number roughly corresponds to the driving hours of one taxi driver in their entire work life. The collected data cover most of the common scenarios, different illumination conditions, and weather, but still not enough to allow completely safe driving [1]. An autonomous road vehicle is expected to encounter a large variety of environmental conditions which might be difficult to take into account fully during the development of its perception modules. Furthermore, the occurrence of specific situations may be rare, and for this reason, hard to grab in a dataset. Increasing the size of the dataset, increases, in turn, the probability of encountering rare events; however, it does not guarantee to assign their proper relevance. Single events may be considered as outliers, and, for this reason, the overall network may perform poorly in such situations.

In this paper, we differentiated the datasets for testing and training our networks by illumination condition (day and night), and by weather conditions (sunny and rain), demonstrating that data fusion techniques and semi-supervised learning may help in segmenting objects in such conditions, showing how the availability of big quantities of data, including non-annotated scenes, can improve the performance of AI-based algorithms. Specifically, it is important to point out that these classes may be strongly unbalanced as sunny days may be better represented in the dataset than rainy nights. For instance, the task of segmenting vehicles or people presents intrinsic difficulties, because all classes of objects during the day, are very well represented in the dataset, whereas some classes, such as people in rainy nights, are more rare (for obvious reasons), but remain important to detect with high confidence.

Our paper thus contributes to the body of knowledge in this field, investigating how sensor fusion and semi-supervised learning can be used to increase the overall performance of the network with a particular emphasis on uncommon events and challenging scenarios. The objective was to carry out a fair evaluation, also using cross validation, of the techniques and how they can be used to improve machine learning in autonomous driving. The study investigated and integrated two lines of research: the use of an individual sensor setup versus sensor fusion, and how to use semi-supervised learning to improve overall network performance using unlabelled data coming from one of the sensors—in this case, the RGB camera or the lidar. It is not in the scope of the present paper to beat the current benchmarking in object segmentation, but to show readers how fusion and semi-supervised learning can be used to improve performance in AI algorithms. However, the results suggest significant segmentation capability improvement in night and in rainy conditions, ranging from 10 to 30 percentage points.

To achieve this goal, we trained different models in a supervised fashion, with data fusion and semi-supervised learning. The supervised learning technique was used to train a baseline model and an upperbound model was used for comparison. The expected result is that the upperbound model would be the best performing one, benefiting from the full knowledge of the full dataset with data annotation. However, acquiring real-world scenarios with full data annotation is not always achievable, especially in tasks such as autonomous driving. Thus, this paper shows that semi-supervised learning and co-training achieve comparable performance (about 2–3 percent points difference) using less data annotations. An additional point of discussion is the cross-validation on different data splits. In this study, we trained 10 different models for each train–validation–test to show the variance in the test results.

This paper is organized as follows: Section 2 introduces the reader to the topic, offering a review of the state of the art, including recent studies about sensor fusion and semi-supervised learning. The materials and datasets used for this study are thoroughly explained in Section 3, including the Waymo dataset used for this research. Section 4 addresses our method for building the neural networks, training, validation and testing modalities. Finally, our results are reported in Section 5 including a comparison with our previous method, and a discussion of our main findings.

## 2. Related Work

Most of the state-of-the art methods for autonomous driving involve data-driven techniques at various levels, among which deep neural networks are shown to be promising in solving scene interpretation problems. Working on improving scene interpretation, this paper is focused on the intersection of two problems: sensor fusion and semi-supervised learning. Both topics have been extensively explored in the literature.

Semi-supervised learning is a widely explored idea for exploiting the availability of big unlabelled datasets to train various types of neural networks. In a recent review [2], Van Engelen et al. explored the topic from a broad non-task-specific perspective. Despite the fact that the idea of semi-supervised learning is applicable in different ways, and for several sources of information, images classification and semantic segmentation are the most historically used. Ouali et al. [3] used labelled data to train the main encoder–decoder-based network for semantic segmentation using the PASCAL VOC dataset [4]. The unlabelled data were used in a second stage to train the same network with the addition of auxiliary decoders, and perform a consistency check between the main decoder and the auxiliary decoders. The potential of semi-supervised learning was also used in [5] to build a network for semantic segmentation where strong pixel-level annotation is only available for part of the dataset, and weak annotation (image-level) is available for the remaining part of the dataset. The semantic segmentation generated on the weakly annotated images was used to train the overall network. Also in this case, the examples were taken from the PASCAL VOC dataset. The main driver for semi-supervised learning was to reduce the cost of labelling [6], which is time-consuming and intensive work.

Focusing on the task of autonomous driving, [7] offered a review of methods and datasets, indicating the increment in labelling efficiency, transfer learning, semi-supervised learning, etc., as open questions for research to leverage lifelong learning by updating networks with continual data collection instead of re-training from scratch. One example of application is provided in [8], where a semi-supervised learning method that uses labelled and unlabelled camera images to improve traffic sign recognition is proposed. The semi-supervised learning is also used in [9], where Zhu et al. define a *teacher model* which is trained in a supervised manner using labelled camera images. Then, the teacher was used to generate labels on an unlabelled dataset which was used to train a *student model*. The authors show that the student model outperformed the teacher model using the data from the Cityscapes [10], CamVid [11] and KITTI [12] datasets.

As autonomous vehicles nowadays are integrated with different sensors, 3D lidar data are also used for semantic segmentation. A review work explores the available datasets and emphasises the importance of the availability of big quantities of labelled data coming from 3D lidar that are expensive to label manually, though strongly needed for autonomous driving [13]. A method to achieve the task of semi-supervised learning using 3D lidar data is described in [14], in which a set of manually labelled data and pairwise constraints are used to achieve an improvement in performance.

In addition to many techniques for extracting relevant information from camera or lidar data individually, data fusion is a growing trend to integrate the information coming from both sensors to improve each other in segmentation performance. In [15], the authors offer a review of different methods for sensor fusion perception in autonomous driving using deep learning techniques, focusing on fusion as a means for solving visual odometry, segmentation, detection and mapping issues, pointing out in their conclusion of how adverse weather can affect overall performance. A focused review on sensors’ performance under adverse weather conditions can be found in [16], in which the authors better describe the individual strengths and limitations of sensors in the automotive field, providing a comprehensive list of data-driven methods and an open dataset. However, the literature is rich in approaches to sensor fusion that use classical stochastic inference instead of neural networks. For instance, in [17], the author generalizes the approach in [18] with the objective to obtain quality-fused values from multiple sources of probabilistic distributions in which quality is related to the lack of uncertainty in the fused value and the use of credible sources. On a different research line, the authors in [19] addressed the problem of sensor fusion and data analysis integration with emerging technologies and described several classic methods for sensor fusion, such as Kalman filtering and Bayesian inference. The strengths of these methods reside in their simplicity and the high level of control they offer over the design process, with the drawback of low flexibility and adaptability. On the contrary, convolutional neural networks have demonstrated high flexibility and adaptability to input variations, with the drawback of losing control over the design process—as CNNs are, essentially, black boxes.

Among many dedicated techniques of lidar camera fusion that can be found in the literature, a relevant example is described in [20], where Li et al. defined the so-called “BiFNet” as a bidirectional network for road segmentation that uses camera image and lidar eye-bird view. In [21], a lidar–camera cross fusion technique was presented, showing an increment in performance using the fusion technique over an individual sensor on the KITTI dataset, and later extended using the co-training method that included labelled and unlabelled examples [22].

## 3. Materials

In this work, we used the Waymo open dataset [23]. This section first presents a general overview of the dataset. Afterwards, we describe the procedure that was used for converting unordered point clouds into images, and for generating semantic masks from 3D bounding boxes.

### 3.1. Waymo Dataset

The Waymo open dataset includes 1110 driving sequences recorded with multiple cameras and lidars across a large variety of locations, road types and weather and lighting conditions. Each driving sequence consists of a 20-s-long recording sampled at 10 Hz. Both 2D and 3D bounding boxes were manually generated for all frames and considering the following four categories of objects: vehicles; pedestrians; cyclists; and traffic signs. Additionally, the driving sequences were partitioned into four broad subsets, namely day–fair; night–fair; day–rain; night–rain (see Table 1 for further details). The labels day and night indicate whether a sequence was collected during the day under good lighting conditions, or late in the day or at night under poor external illumination. The labels fair and rain instead refer to the weather conditions, with fair denoting good weather, and rain denoting active raining or wet environment following recent precipitation.

### 3.2. Point Cloud Projection

The literature is rich in approaches to process point clouds with deep neural networks, see for example [24,25]. In this work, the lidar point cloud is simply projected into the camera plane in order to generate a three-channel tensor with the same width and height of the RGB image, and such that each channel encodes one of the 3D spatial coordinates [21]. By doing so, it is straightforward to establish a one-to-one correspondence between the colour information, contained in the RGB image, and the spatial information, contained in the point cloud. A point cloud acquired with a Velodyne HDL-64E consists of approximately 100,000 points where each point *p* is specified by its spatial coordinates in the lidar coordinate system, that is $p={\left[x,y,z,1\right]}^{T}$. Given the lidar–camera transformation matrix T, the rectification matrix R, and the camera projection matrix P, it is possible to calculate the column position *u*, the row position *v*, and the scaling factor α, where the projection of *p* intersects the camera plane, by solving the following expression $\alpha \;{\left[u,v,1\right]}^{T}=P\;R\;T\;p$.This procedure is applied to every point in the point cloud, while discarding points such that α<0 or when [u,v] falls outside the image. By using the above procedure, three images denoted as X, Y and Z are generated where each pixel contains the *x*, *y*, and *z* coordinates of the 3D point that was projected into it.

### 3.3. Sparse Semantic Masks from 3D Bounding Boxes

As mentioned in Section 3.1, the annotations provided for the Waymo dataset are 2D and 3D bounding boxes. Here, however, we are interested in carrying out semantic segmentation; for this reason, the 3D bounding boxes are converted into semantic masks. This can be easily achieved by using a procedure analogous to the point cloud projection mentioned in Section 3.2. More specifically, given a 3D bounding box, we collect all the lidar points that fall within it, and then project them into the image plane. Each projected point is drawn in the semantic mask as a disk with the same class as the bounding box. This procedure is repeated for all the 3D bounding boxes found in a given frame. In this work, only the vehicle class is considered which is most represented and evenly distributed across the coarse categories described in Section 3.1. Some examples of semantic masks obtained using this method are shown in Figure 1. A limitation of this procedure is that only regions of an image where there are lidar detections can be assigned to a valid class. All remaining pixels are assigned to a do-not-care class that is ignored during training. As illustrated in the bottom three rows of Figure 1, poor illumination and rainy weather might affect the quality of the sensor data which could have detrimental effects for downstream applications.

## 4. Method

This section describes the methodology proposed to provide a quantitative evaluation and a comparison of semi-supervised learning techniques against a baseline network and sensor fusion methodologies, including the models’ design, data splits used for the cross validation and the training procedures used in this paper.

### 4.1. Model

The base network architecture used in this work is the well-known FCN-ResNet50 [26]. This CNN contains five stages, denoted as S1–S5, where each stage consists of several layers (e.g., convolutional, batch normalization, max-pooling, etc.). The proposed model contains three subnetworks, namely RGB, Lidar and Fusion, and it is shown in Figure 2. The RGB and Lidar subnetworks have the same structure as the base FCN-ResNet50 and as described by their names, receive as input camera images and lidar images, respectively. The fusion subnetwork instead processes the concatenated features of the single modality branches after stage 4. This can be described as a late fusion strategy [21]. As illustrated in Figure 2, the fusion subnetwork shares some of its stages (S1–S4) with the single modality subnetworks. All networks’ weights were initialized with an FCN-ResNet50 pretrained on a subset of COCO train2017.

### 4.2. Dataset Splits

As mentioned in Section 3.1, in this work, we used the Waymo dataset which consists of 1110 driving sequences collected under various weather and lighting conditions. Each sequence originally contains approximately 200 frames (i.e., 20 s sampled at 10 Hz) of which we kept every 10th frame. Out of the full dataset, for the experiments described in the following sections, we generated N=10 random dataset splits $S_{i}=\left\{T_{i},V_{i},U_{i},K_{i}\right\}$, where i=1,…,N. The randomization was carried out by considering driving sequences instead of individual frames in order to avoid any overlap between the subsets in any given $S_{i}$. Subset $T_{i}$ denotes a training set, which is a collection of annotated examples used for computing the loss function and updating the model parameters. Subset $V_{i}$ denotes a validation set which is used to carry out early stopping and hyper-parameter tuning. Subset $U_{i}$ denotes a collection of examples for semi-supervised learning. That is, examples that are assumed to be unlabelled and for which the models should generate proxy labels. Lastly, each $K_{i}$ denotes a test set used for evaluating the generalization performance of the model for unseen data. With the exception of the training sets T, which only contain examples belonging to the coarse category day–fair, all other sets include examples belonging to all coarse categories. By only considering training data belonging to one category, it is possible to investigate the effectiveness of the proposed approach for carrying out domain adaptation. Table 2 provides more details regarding the dataset splits S.

### 4.3. Training

The training procedure was designed to iterate two phases for any given dataset split $S_{i}$.

**Phase 1:** The model is trained in a purely supervised fashion with training set $T_{i}$ and validation set $V_{i}$. The intersection over union (IoU) is computed on the validation set at the end of each epoch for all the subnetworks. If any of the subnetworks’ IoU has improved, a copy of the complete model is stored. The IoU index is commonly used for segmentation tasks in deep learning [27], and it is calculated as follows:(1)$$IoU=\frac{A_{T}\cap A_{P}}{A_{T}\cup A_{P}},$$
where $A_{T}$ is the ground truth semantic area of a specific object, and $A_{P}$ is the area of the object predicted by the network. This index is maximized when the union is equal to the intersection.

**Phase 2:** The model containing the best performing fusion subnetwork, denoted as $\psi_{i}$, is used as starting point for training four different models:A supervised-learning baseline model;A model trained using a semi-supervised approach called co-training [22];A model trained using a newly introduced sensor fusion-based semi-supervised approach;A supervised-learning upperbound model.

For both phases, we used the cross-entropy loss function and Adam optimization [28]. The batch size was set to 32. The learning rate was decayed using the poly-learning policy [29] implemented as
(2)$$\eta \left(i\right)=\eta_{0}{\left(1-\frac{j}{N}\right)}^{\alpha },$$
where *j* denotes the current epoch number, $\eta_{0}$ is the starting learning rate, *N* represents the total number of the training epoch and α=0.9. In the first phase, we set $\eta_{0}=0.0003$ and N=50, whereas in the second phase, we set $\eta_{0}=0.0001$ and N=100. The values indicated above were empirically determined after a hyper-parameter search and fixed for all the trainings, though the perfect optimization of these parameters is beyond the scope of the paper. The images were first downsampled from 1920×1280 to 480×320 pixels. Afterwards, the top half region, usually containing buildings, vegetation and the sky was discarded. The input images thus had a size of 240×320 pixels. Data augmentation consisted of three operations, specifically random square cropping, random rotations in the range $\left[-20^{\circ },20^{\circ }\right]$ about the centre of the images, and random colour jittering (brightness, contrast, saturation and hue) applied to the RGB images.

### 4.4. Supervised Learning Baseline

Given a model ψ, the three subnetworks are denoted as $\psi^{\beta }\equiv \psi^{\beta }\left(\theta_{\beta }\right)$, where β∈{rgb,lid,fus} and $\theta_{\beta }$ represent the subnetwork’s weights. For ease of notation, in the following discussion, we omit the weight vector from the equations. As mentioned in Section 4.1, some of weights are shared between the subnetworks, therefore $\theta_{\mathrm{lid},\mathrm{rgb}}\cap \theta_{\mathrm{fus}}\neq Ø$. An input example has two views, lidar and camera, and is denoted as $x=\{x_{\mathrm{rgb}},x_{\mathrm{lid}}\}$. The corresponding ground truth semantic mask is denoted as *y*. The cross entropy between a subnetwork’s prediction and the ground truth is represented as $H(\psi^{\beta }\left(x\right),y)$. In the previous expression, we assumed that each subnetwork extracts the appropriate input view, that is, $\psi^{\beta }\left(x\right)\equiv \psi^{\beta }\left(x_{\beta }\right)$. The total loss function used for supervised learning is then given by the following expression:(3)$$L_{\sup}=\sum_{\beta \in \{\mathrm{rgb},\mathrm{lid},\mathrm{fus}\}}E_{(x,y)\in T}\left[H\left(\psi^{\beta }\left(x\right),y\right)\right].$$

As indicated in Phase 1 of Section 4.3, the baseline model is trained in a supervised fashion using the loss function indicated in Equation (Equation 3). The backbone of our networks is always a pretrained FCN-ResNet50.

### 4.5. Co-Training

Co-training is a semi-supervised learning algorithm that can be applied to problems where the *instance space* can be partitioned into two independent *views*. The instance space is an abstraction of the input space associated with a classification problem, whereas the views contain the actual data that will be consumed by the classifiers. The predictions obtained in one view can be employed as labels in the other view with the final goal of leverage for boosting performance. This approach has two strengths: (1) reduced effort in manual annotation; and (2) increased knowledge learning from the independent views. In this work, following a previous line of research described in [22], the instance space consists of urban driving scenes, and the views are provided in the form of RGB images and lidar point clouds. The predictions of a lidar-based semantic segmentation network, which is generally less affected by environment illumination (day/night light), could be exploited by a camera-based network in order to learn more discriminative features in challenging conditions such as rainy days or night light.

Let us consider a teacher network $\psi_{t}$ and a student network $\psi_{s}$, parametrized by weights $\theta_{t}$ and $\theta_{t}$, respectively, and their relative views $\left(x_{t},x_{s},y\right)\in T_{i}$ that correspond to the teacher view $x_{t}$, the student view $x_{s}$, the ground truth labels *y* and the training set for any given data split $T_{i}$. The fundamental building block for the loss function is the well-known cross-entropy loss, denoted as *H*, between the student prediction and the ground truth label. The student loss is calculated using the supervised loss in Equation (Equation 3) for each data view (RGB, lidar). For an unlabelled example in $U_{i}$, the student’s prediction and the teacher’s prediction represent probability distributions over possible classes, though the teacher’s is considered as the ground truth to train the student model. By considering that the Kullback–Leibler divergence [30], denoted as $D_{\mathrm{KL}}$ is a measure of difference between probability distributions, the co-training loss is implemented as follows:(4)$$L_{\cot}=L_{\sup}+E_{(x,y)\in U}[D_{\mathrm{KL}}(\psi^{\mathrm{rgb}}\left(x\right)||\psi^{\mathrm{lid}}\left(x\right))+D_{\mathrm{KL}}(\psi^{\mathrm{lid}}\left(x\right)||\psi^{\mathrm{rgb}}\left(x\right))].$$

### 4.6. Fusion Proxy Labels

Typically, in co-training, the proxy labels are obtained using the single modality subnetworks. However, it has been shown that a model generated using data fusion typically achieves higher performance [21]. For this reason, it is expected that the proxy labels obtained by the fusion subnetwork will be more reliable. The agreement between fusion network and lidar/camera subnetworks is performed for each subnetwork using the $D_{\mathrm{KL}}$ as follows:(5)$$L_{\mathrm{fus}}=L_{\sup}+E_{(x,y)\in U}[D_{\mathrm{KL}}(\psi^{\mathrm{fus}}\left(x\right)||\psi^{\mathrm{lid}}\left(x\right))+D_{\mathrm{KL}}(\psi^{\mathrm{fus}}\left(x\right)||\psi^{\mathrm{rgb}}\left(x\right))].$$

## 5. Results and Discussion

The model detailed in Section 4.1 was trained on the 10 random dataset splits $S_{i}$, described in Section 4.2, and extracted from the Waymo dataset (see Section 3.1 for more details). The full training procedure is outlined in Section 4.

Furthermore, training the model more times provides a reliable result in terms of repeatability, as individual training bouts may perform differently according to different conditions in data-loading randomization, dropout layers and weights initialization. The results on the test sets are summarized in Table 3 and in Figure 3.

### 5.1. Supervised Baseline

The supervised baseline models were trained as described in Section 4.4 using only sequences collected in day-time and fair weather conditions. As can be seen in Table 3, the fusion subnetwork performs significantly better than the single modality subnetworks in both the day–fair and day–rain categories. However, the lidar subnetwork performs best in the night-time categories where the performance of the camera subnetwork drops significantly. The fact that the lidar subnetwork performs better than the camera one at night-time is not surprising considering that the lidar is an active sensor. The fusion subnetwork also has access to the lidar input, so its poor performance in night-time sequences is an indication that, during training, it has learned to rely too strongly on the camera-based features. Another interesting result is that the lidar subnetwork performance is negatively affected by rainy weather. As shown in two examples in Figure 1, the water covering the surrounding surfaces degrades the density, and possibly the quality, of the point clouds captured with the lidar.

### 5.2. Co-Training

Co-training shows slight improvements for day–fair weather data, which is expected as full information content already provides reliable results for the supervised baseline. The improvement using the co-training approach with respect to the baseline is higher in case of night–fair data than day–light data. Here, the performance is over 80% accuracy in almost all generated models, showing an improvement in the camera-based network case of about 15 percent points. The only case in which the performance is lower, but still a big improvement with respect to camera data baseline, is the night–rain scenario, in which the performance against the baseline increases in all cases ranging between 5 percent points, in the lidar case, to almost 30 points, in the camera case.

### 5.3. Fusion-Based Semi-Supervised Learning

Result from Table 3 for semi-supervised fusion and co-training are fully comparable, with only little difference in all scenarios. In most of the cases, fusion seems to outperform the co-training on average, with some cases in which the opposite happens. The variance, however, shows that co-training seems to be more stable, showing lower variance in many cases. In all cases, there is an improvement that is more significant in the case of night and rainy weather and even comparable to the upperbound.

An important observation is that the upperbound model clearly shows the best performance in all cases, which is not a surprise. It is well known that the availability of large quantities of well-annotated data is a fundamental issue, and the best performance is achieved with increased information availability. However, real-world cases show that this is not always possible, achievable or cost-effective. This result shows that co-training and semi-supervised learning can help to fill this gap, and in some cases even over-performing on the upperbound model, for example, in the case of night–fair lidar data. It is reasonable to expected that semi-supervised learning and co-training would surpass the upperbound model if more data were available.

### 5.4. Cross-Validation

The results in Table 3 also show the variance across several models trained under the same conditions but on different data splits (as described in Section 4.1). Using cross-validation, these results confirm that fusion and co-training have similar performance improving the baseline performance, reaching the upperbound-level performance. Figure 3b shows the performance for each subnetwork (RGB, lidar and fusion) of each data split, the average accuracy and the variance. From Figure 3a, one can observe that in the day–fair case, the camera performances are comparable in all cases (baseline, co-training, fusion and upperboud), while fusion and co-training clearly help in rain and night for all data splits. The same behaviour can be seen in Figure 3b for the lidar sub-network, though showing (as expected) better performance during night-time. Finally, Figure 3c shows the performance distribution over all the data splits for the fusion sub-network.

### 5.5. Discussion

The purpose of the experiments proposed in Section 5 is three-fold: (1) to evaluate the potential benefits of using a multi-sensor system over single modality approaches; (2) to investigate the model generalization capabilities to more challenging domains; and (3) to evaluate whether co-training and our newly introduced fusion-based strategy could be useful for domain adaptation.

According to the results shown in Section 5, several potential benefits support the use of semi-supervised learning, sensor fusion and co-training strategies. First, a multi-sensor system is shown to provide high reliability and redundancy, for all the cases reported in Table 3 in which sensor-fusion outperforms the single-sensor approach (RGB camera or lidar). Second, Table 3 also shows that co-training and semi-supervised learning help the model to generalize better to more challenging domains. The best improvement in the semi-supervised learning techniques is shown in the night–rain case with an over 10 percent points IoU performance increase, starting from 70% IoU accuracy of the fusion baseline, and reaching 81% in the fusion-semi case.

The use of non-annotated data has clearly been shown to improve the overall performance, resulting in big savings in practical applications where data annotation is a heavy and complex task. However, data annotation cannot be neglected, the upperbound model is shown to be the best performing one, and hence full data annotation is still the best way to achieve high reliability and stability in neural networks.

## 6. Conclusions and Future Work

In conclusion, this paper offered a comparative study that analysed two semi-supervised methods of sensor fusion techniques for lidar–camera data in deep learning, showing a comparison among different networks’ performance, the baseline model and a supervised upperbound model. Our results confirm the overall trend that the semi-supervised method could boost performance, taking advantage of the availability of a big un-annotated dataset. The paper shows that the upperbound model performance level can be reached using other methods such as semi-supervised learning and co-training, resulting in a cost-effective method that uses less data annotation. This result is supported by a cross-validation using 10 different data splits. Furthermore, the statistical analysis on single model training shows how benchmarking in autonomous driving could be affected by randomization in individual training.

Future work could extend the analysis over different conditions including additional subcategories in which deep learning in autonomous driving performance still suffers from the availability of data such as adverse weather-related data.

## Figures and Tables

**Figure 1 sensors-21-04813-f001:**
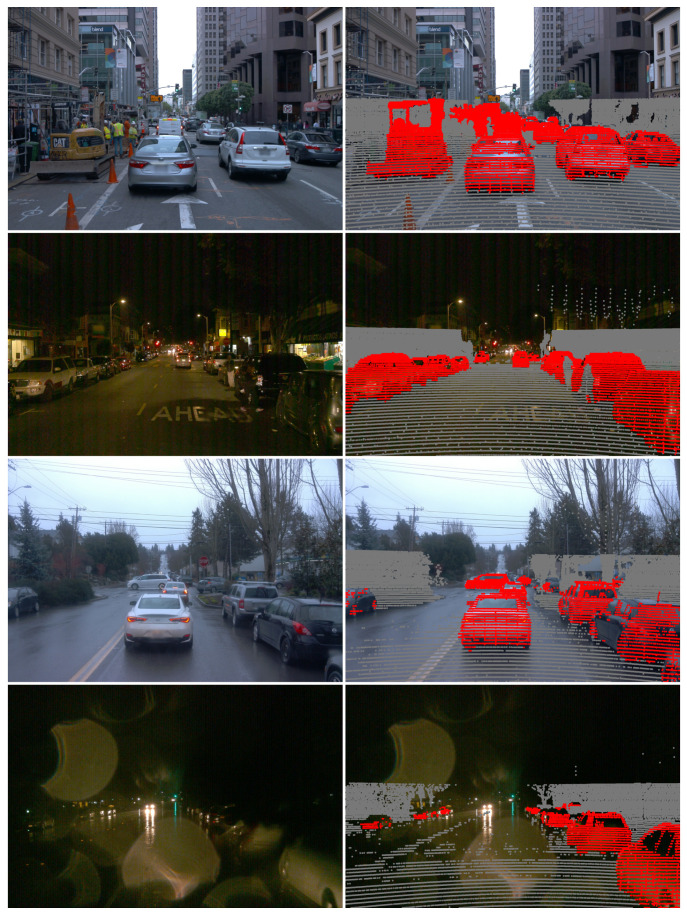
Some examples of driving scenes from the Waymo dataset. The left column shows four RGB images captured under various lighting and weather conditions, whereas the right column contains the corresponding semantic masks obtained using the procedure described in Section 3.3. Red pixels denote the vehicle class, while grey pixels denote the background (i.e., negative class). All other pixels in the semantic mask are ignored during training.

**Figure 2 sensors-21-04813-f002:**
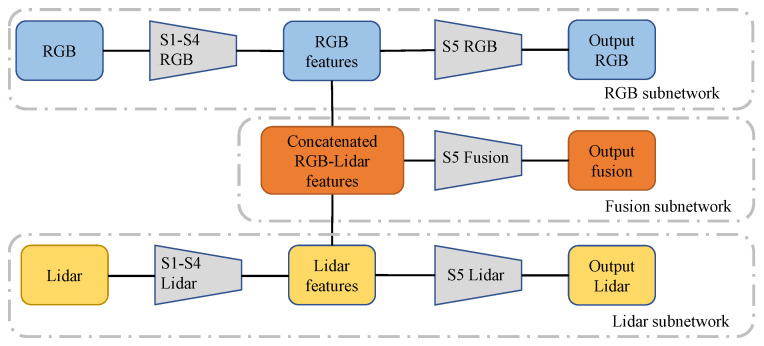
Illustration of the model used in this work. The model can be decomposed into three subnetworks: RGB; lidar; and fusion. The fusion subnetwork receives as input the concatenated features generated by stages S1–S4 of the single modality subnetworks. This corresponds to a late fusion strategy. During a supervised learning step, the output of each subnetwork is used to compute a loss term with respect to a manually generated ground truth. During a semi-supervised step, the fusion subnetwork acts as teacher for the single modality ones. That is, the fusion subnetwork’s output is used as ground truth for computing the losses of the RGB and lidar subnetworks. For co-training, only the single modality subnetworks are considered. In this case, the output relative to one sensor is used as ground truth for the other.

**Figure 3 sensors-21-04813-f003:**
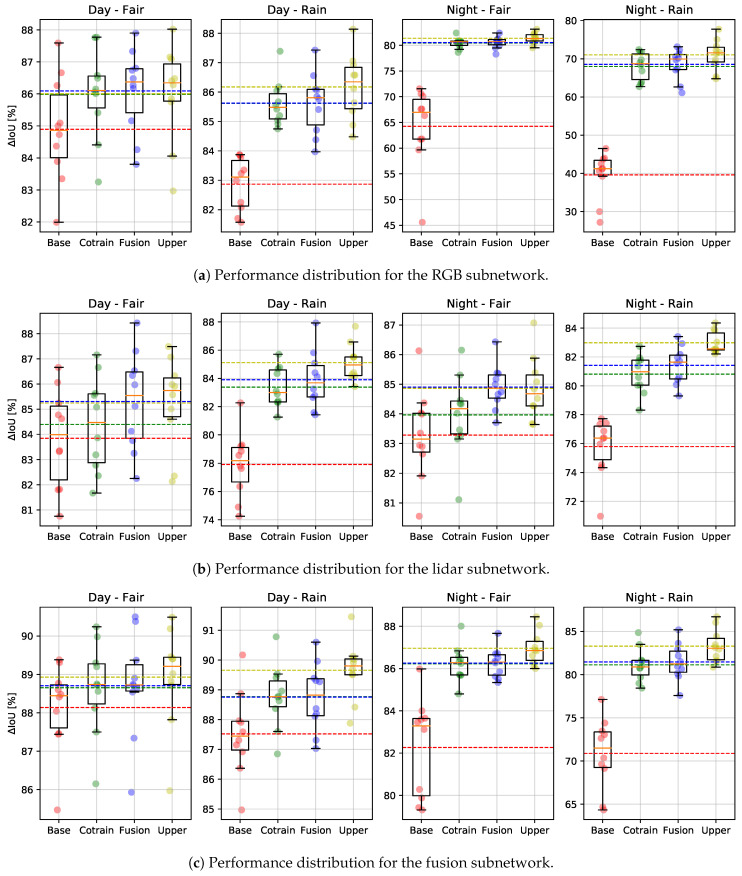
Performance distribution over the networks’ cross-validation on data splits; a detailed description can be seen in Section 5.4.

**Table 1 sensors-21-04813-t001:** Number of 20-s-long driving sequences belonging to the four main categories considered in this work (10 Hz sample frequency).

Day–Fair	Day–Rain	Night–Fair	Night–Rain
747	226	82	45

**Table 2 sensors-21-04813-t002:** Number of sequences assigned to training, validation, unlabelled, and test subsets for each dataset split $S_{i}$ according to the four main categories considered in this work.

Set	Day-Fair	Day-Rain	Night-Fair	Night-Rain
Training (T)	100	0	0	0
Validation (V)	10	10	10	5
Unlabeled (U)	40	40	36	20
Test (K)	40	40	36	20

**Table 3 sensors-21-04813-t003:** Main results showing the vehicle class average IoU in percentage, and standard deviation on the test sets (see Section 4.2). The numbers within parentheses denote the average improvement with respect to the supervised baseline.

Category	Modality	Baseline (%)	Co-Training (%)	Fusion-Semi (%)	Upperbound (%)
Day–fair	camera	84.89±1.56	85.99±1.31(1.10)	86.09±1.24(1.20)	86.01±1.42(1.11)
	lidar	83.84±1.86	84.39±1.79(0.56)	85.31±1.85(1.47)	85.24±1.71(1.41)
	fusion	88.14±1.09	88.65±1.14(0.52)	88.71±1.27(0.57)	88.93±1.22(0.79)
Day–rain	camera	82.87±0.85	85.62±0.74(2.75)	85.62±0.99(2.76)	86.18±1.05(3.31)
	lidar	77.91±2.21	83.38±1.34(5.47)	83.90±1.91(5.99)	85.11±1.24(7.20)
	fusion	87.52±1.33	88.77±1.01(1.25)	88.76±1.08(1.23)	89.66±0.92(2.13)
Night–fair	camera	64.25±7.32	80.43±0.98(16.18)	80.53±1.05(16.29)	81.36±1.02(17.11)
	lidar	83.28±1.43	83.96±1.29(0.68)	84.90±0.72(1.62)	84.86±1.00(1.58)
	fusion	82.27±2.22	86.23±0.81(3.96)	86.26±0.67(3.99)	86.96±0.76(4.69)
Night–rain	camera	39.61±5.87	67.99±3.47(28.38)	68.54±3.80(28.93)	71.02±3.82(31.41)
	lidar	75.78±1.93	80.81±1.28(5.02)	81.42±1.23(5.63)	82.98±0.74(7.20)
	fusion	70.88±1.33	81.13±1.01(10.25)	81.46±1.08(10.58)	83.29±0.92(12.41)

## Data Availability

The data used in this research is available on github at https://github.com/bellonemauro/LCSSLSS-DataSplits (accessed on 14 July 2021).

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
