# Peer review of "Lidar–Camera Semi-Supervised Learning for Semantic Segmentation"

_sensors, 2021, doi:10.3390/s21144813_

Round 1

Reviewer 1 Report

In this paper, the authors improve the overall performance of the network by using sensor fusion and semi-supervised learning. The work is very interesting and meaningful. But there are some problems that the authors need to pay attention to, as follows:

1. The data sources in Section 5 is suggested to provide.

2. The advantages of the proposed method is suggested to discuss in the Section 5.

3. Please explain how the values of each variable are determined in Eq. (1) in detail.

4. The logic between each Section needs to be improved.

5. In Section 4.3, how to calculate IoU should be explained in detail in the form of the mathematical formula. Otherwise, it is different to understand the results in Fig. 3.

6. In Section 5.1, what the baseline models should be explained in detail in the form of the mathematical formula. It is suggested to add corresponding references. If it is related to Section 4.1, the expression should be consistent. Otherwise, it confuses readers.

7. In Table 3, % should be added to keep the consistency.

8. For sensor data fusion, there are many other current related works, e.g,

1) GIQ: A generalized intelligent quality-based approach for fusing multi-source information;

2)  An overview of IoT sensor data processing, fusion, and analysis techniques.

It is suggested to discuss them in the background, and clarify why selecting semi-supervised learning to handle sensor fusion.

In summary, this article has practical value and research significance. I recommend accepting this paper after addressing the above comments.

Author Response

Reviewer #1:

In this paper, the authors improve the overall performance of the network by using sensor fusion and semi-supervised learning. The work is very interesting and meaningful. But there are some problems that the authors need to pay attention to, as follows:

Reply:

Many thanks for carefully reading our paper and giving many valuable comments. We have now revised, taking into account your comments and suggestions to the best of our ability, and we hope that you will find the revised version acceptable.

Comment #1: The data sources in Section 5 is suggested to provide.

Reply: The data source was described in Section 3.1, though missing in Section 5 as pointed out by the reviewer. We have changed the text to clarify this in the introduction of Section 5.

Action:

Previous lines 243-245: reading:

… “Towards that end, the model detailed in Section 4.1 was trained on the 10 random data set splits S_i described in Section 4.2 using the training procedure previously outlined” …

are now revised to

… “Towards that end, the model detailed in Section 4.1 was trained on the 10 random data set splits S_i, described in Section 4.2, and extracted from the Waymo dataset (see Section 3.1 for more details). The full training procedure is outlined in Section 4.”

Comment #2: The advantage of the proposed method is suggested to discuss in Section 5.

Reply: Thanks for this comment; a more detailed description of the strengths of this method should help readers to better appreciate the value of this work.

Action: An additional discussion of the strengths of our method is now added to the end of Section 5, now reading:

“The purpose of the experiments proposed in Section 5 was three-fold: (1) to evaluate the potential benefits of using a multi-sensor system over single modality approaches, (2) to investigate the model generalization capabilities to more challenging domains, and (3) to evaluate whether co-training and our newly introduced fusion-based strategy could be useful for domain adaptation.

According to the results shown in Section 5, there are several potential benefits supporting the use of semi-supervised learning, sensor fusion and co-training strategies. First, a multi-sensor system is shown to provide high reliability and redundancy: for all the cases reported in Table 3 sensor-fusion outperforms the single sensor approach (RGB camera or lidar). Second, Table 3 also shows that co-training and semi-supervised learning help the model to generalize better to more challenging domains. In fact, the best improvement of semi-supervised learning techniques is shown in the night-rain case with over 10% IoU performance increase, starting from 70% of the fusion baseline and reaching 81% in the fusion-semi case.

The use of non-annotated data has clearly shown to improve the overall performance, resulting in big savings in practical applications where data annotation is a heavy and complex task. However, data annotation cannot be neglected, the upperbound model is shown to be the best performing one, hence full data annotation is still the best way to achieve high reliability and stability in neural networks.”

Comment #3: Please explain how the values of each variable are determined in Eq. (1) in detail.

Reply: Parameters such as base learning rate and learning decay are estimated empirically and deducted after a hyperparameter search. As the main aim of the paper was not to beat the benchmark but to demonstrate how semi-supervised learning could be used in this task, we have not optimized those parameters but rather used classical values that worked well for our application. A short explanation about this is now added after Eq. (1).

Action:

The following text was added to Section 4.3:

“The values indicated above were determined empirically after a hyper-parameter search and fixed for all the trainings, though the perfect optimization of these parameters is beyond the scope of the paper.” 

Comment #4: The logic between each Section needs to be improved.

Reply: We agree with the reviewer, particularly in connection with the description of the baseline model and supervised learning there was some possible misinterpretation. We have made many changes during this review process, including reorganization of sections, which we believe have improved the general quality of this paper. The change list of the reorganization of sections is listed below, whereas specific sentences are included in other answers to this reviewer’s comments thus not mentioned here for brevity.

Action:

Section 4.4 – “Supervised learning” renamed as “Supervised learning baseline”.

Subsection 5.1 – “Baseline” renamed as “Supervised baseline”.

Subsection 5.5 – “Discussion” was added to the paper.

Comment #5: In Section 4.3, how to calculate IoU should be explained in detail in the form of the mathematical formula. Otherwise, it is different to understand the results in Fig. 3.

Reply: Thanks to the reviewer’s comment we have now added the IoU formula and a better explanation in Section 4.3.

Action:

The following text is added to Section 4.3:

“The IoU index is commonly used for segmentation tasks in deep learning [27], and it is calculated as follows:

where  is the ground truth semantic area of a specific object, and  is the area of the object predicted by the network. This index is maximized when the union is equal to the intersection.”

Comment #6: In Section 5.1, what the baseline models should be explained in detail in the form of the mathematical formula. It is suggested to add corresponding references. If it is related to Section 4.1, the expression should be consistent. Otherwise, it confuses readers.

Reply: The backbone of our network is the fully convolutional neural network ResNet50, this was explained in Section 4.1 – “Model”. However, we agree that to improve clarity some further clarification was needed. For this reason, we have changed Section 4.4 to better explain that supervised learning was referring to our baseline model by adding an additional paragraph. Furthermore, in Section 5, the title of the subsection 5.1 – Baseline is now changed to “Supervised baseline” adding the corresponding references to the specific sections.

Action:

The following text was added in Section 4.4:

“As indicated in Phase 1 of Section 4.3, the baseline model is trained in a supervised fashion using the loss function indicated in (3). The backbone of our networks is always a pretrained FCN-ResNet50.”

The title of Section 5.1 was changed from “baseline” to “supervised baseline” to improve consistency and clarity.

Previous line 250 in Section 5.1

          “The baseline models were trained using only sequences collected …”

was changed, now reading:

“The supervised baseline models were trained as described in Section 4.4 using only sequences collected …”

Comment #7: In Table 3, % should be added to keep the consistency.

Reply: Thanks to the reviewer for noting this, the symbol is now added to the table.

Action: Table 3 was fixed.

Comment #8: For sensor data fusion, there are many other current related works, e.g,

1) GIQ: A generalized intelligent quality-based approach for fusing multi-source information;

2)  An overview of IoT sensor data processing, fusion, and analysis techniques.

It is suggested to discuss them in the background, and clarify why selecting semi-supervised learning to handle sensor fusion.

In summary, this article has practical value and research significance. I recommend accepting this paper after addressing the above comments.

Reply: The reviewer pointed out references describing sensor fusion approaches that are not based on CNNs, but rather on classical probability methods. The value of these methods resides in the high level of control over the algorithms, while lacking flexibility to changing situations. These methods are typically based on the estimation of a weighted function to perform sensor fusion that is fixed once and for all.

In the field of autonomous driving sensors have different reliability in different conditions and we expect the network to adapt to data-reliability according to different conditions (for example lidar is supposed to be more reliable in the night, while camera works fairly well on the day). In such a condition, information coming from different sources should be weighted differently, and neural network are good in doing exactly this, reason behind our choice.

We have now discussed these references on the related work section.  

Action:

The following text is now added to Section 2.

… “However, the literature is rich in approaches to sensor fusion that use classical stochastic inference instead of neural networks. For instance, in [17] the author generalizes the approach in [18] with the objective to obtain quality-fused values from multiple sources of probabilistic distributions in which quality is related to the lack of uncertainty in the fused value and the use of credible sources. On a different research line, in [19] the authors address the problem of sensor fusion and data analysis integration with emerging technologies and describe several classic methods for sensor fusion such as Kalman filtering and Bayesian inference.

The strengths of these methods reside in their simplicity and high level of control over the design process, with the drawback of low flexibility and adaptability. On the contrary, convolutional neural networks have demonstrated high flexibility and adaptability to input variations, with the drawback of losing control over the design process, CNNs are, essentially, black boxes.” …

Reviewer 2 Report

This paper has provided a study on how lidar-camera fusion can improve semi-supervised semantic segmentation in autonomous driving scenarios. The paper has provided experimental results, which suggest that semi-supervised learning and fusion techniques increase the overall performance of the network in challenging (e.g., from sunny days to rainy nights) scenes.

However, the major problem of this paper is that no new methods are proposed to solve the problem of lidar-camera semi-supervised semantic segmentation. This paper is more like an experimental evaluation of existing methods in different setups. For example, Section 5 has provided the vehicle class average IoU results of different methods, such as Baseline, Co-training, Fusion-semi and Upperbound. But these methods are not new. Another problem is that the abstract of this paper is not quite appropriate, as the abstract is supposed to describe contributions rather than investigations.  Besides, the experimental results are insufficient, more results and figures should be provided.

Author Response

Reviewer #2:

This paper has provided a study on how lidar-camera fusion can improve semi-supervised semantic segmentation in autonomous driving scenarios. The paper has provided experimental results, which suggest that semi-supervised learning and fusion techniques increase the overall performance of the network in challenging (e.g., from sunny days to rainy nights) scenes.

However, the major problem of this paper is that no new methods are proposed to solve the problem of lidar-camera semi-supervised semantic segmentation. This paper is more like an experimental evaluation of existing methods in different setups. For example, Section 5 has provided the vehicle class average IoU results of different methods, such as Baseline, Co-training, Fusion-semi and Upperbound. But these methods are not new. Another problem is that the abstract of this paper is not quite appropriate, as the abstract is supposed to describe contributions rather than investigations.  Besides, the experimental results are insufficient, more results and figures should be provided.

Reply: This paper investigates how the studied methods of semi-supervised learning, co-training and sensor fusion, referenced in the related works, could be applied to autonomous driving by giving a quantitative evaluation of how much sensor fusion and co-training could help in challenging conditions (rain and night). To the knowledge of the authors, these methods have had only limited applications in this context in the literature, particularly in terms of generalization of knowledge acquired in common domains (such as sunny days) to more challenging domains (such as night and rain). The paper shows how we can achieve similar performance to the upperbound model using less data annotations. Moreover, this paper provides a comparative study of such methodologies, and we have better addressed this in the abstract.

The scope of the paper is (1) to quantify the potential benefits of using a multi-sensor system over single modality approaches, (2) to investigate the model generalization capabilities to more challenging domains, and (3) to evaluate whether co-training and our newly introduced fusion-based strategy could be useful for domain adaptation, this was also better underlined in a new subsection 5.5 – “Discussion” and the conclusions.

We hope that the reviewer can find this as acceptable.

Actions:

The abstract was changed to underline the approach of the paper as a “comparative study”.

Previous lines 5-6 reading:

“An experimental evaluation on networks trained in different setups has been carried out using various scenarios from sunny days to rainy nights scenes.”

Are now changed to:

“A comparative study has been carried out by providing an experimental evaluation on networks trained in different setups using various scenarios from sunny days to rainy nights scenes.”

Previous lines 9 reading:

“Our results suggest that semi-supervised learning and fusion techniques increase the overall performance of the network in challenging scenarios.”

Is now changed to:

“Our results suggest that semi-supervised learning and fusion techniques increase the overall performance of the network in challenging scenarios using less data annotations.”

The following text is now added to the Conclusions:

“The paper shows that the upperbound model performance level can be reached using other methods such as semi-supervised learning and co-training, resulting in a cost-effective method that uses less data annotation. This result is supported by a cross-validation using 10 different data splits. “

A new Subsection 5.5 – Discussion was added to the paper to better highlight the results of this study and the benefits of the applied methodology, now reading:

“The purpose of the experiments proposed in Section 5 is three-fold: (1) to evaluate the potential benefits of using a multi-sensor system over single modality approaches, (2) to investigate the model generalization capabilities to more challenging domains, and (3) to evaluate whether co-training and our newly introduced fusion-based strategy could be useful for domain adaptation.

According to the results shown in Section 5, there are several potential benefits supporting the use of semi-supervised learning, sensor fusion and co-training strategies.

Firstly, a multi-sensor system is shown to provide high reliability and redundancy, for all the cases reported in Table 3 sensor-fusion over performs the single sensor approach (RGB camera or lidar).

Secondly, Table 3 also shows that co-training and semi-supervised learning help the model to generalize knowledge to more challenging domains.

In fact, the best improvement of semi-supervised learning techniques is shown in the night-rain case with over 10% IoU performance increase, starting from 70% of the fusion baseline and reaching 81% in the fusion-semi case.

The use of non-annotated data has clearly shown to improve the overall performance, resulting in big savings in practical applications where data annotation is a heavy and complex task.

However, data annotation cannot be neglected, the upperbound model is shown to be the best performing one, hence full data annotation is still the best way to achieve high reliability and stability in neural networks.”

Reviewer 3 Report

The authors proposed and developed a semantic segmentation network for semi-supervised learning to address the performance and large labeled data requirements of supervised models. They carried experiments in four different setups, i.e., supervised-learning "baseline" and "upper bound" models, semi-supervised model "co-training," and their proposed semi-supervised "Fusion" approach. The authors trained their model on Waymos open dataset. Their proposed model includes three subnetworks, i.e., RGB, Lidar, and Fusion. The Fusion subnetwork receives the concatenated RGB and Lidar features as input. 

The paper has a good structure with six sections as follows: "1. Introduction ", "2. Related Work", "3. Materials", "4. Method", "5. Results and Discussion", and "6. Conclusions and Future Work".  The authors have also included a good number of references to support their paper.

The conclusion doesn't clearly support the results shown, i.e., the stated boost in performance of the proposed semi-supervised method compared to the supervised upper bound model, and although the feasibility and cost-effectiveness are mentioned in the results section, the reviewer suggests a more detailed discussion of the proposed algorithm performance and trade-offs as well as current proposed model challenges in the conclusions section as well.

Author Response

Reviewer #3:

The authors proposed and developed a semantic segmentation network for semi-supervised learning to address the performance and large labeled data requirements of supervised models. They carried experiments in four different setups, i.e., supervised-learning "baseline" and "upper bound" models, semi-supervised model "co-training," and their proposed semi-supervised "Fusion" approach. The authors trained their model on Waymos open dataset. Their proposed model includes three subnetworks, i.e., RGB, Lidar, and Fusion. The Fusion subnetwork receives the concatenated RGB and Lidar features as input.

The paper has a good structure with six sections as follows: "1. Introduction ", "2. Related Work", "3. Materials", "4. Method", "5. Results and Discussion", and "6. Conclusions and Future Work".  The authors have also included a good number of references to support their paper.

The conclusion doesn't clearly support the results shown, i.e., the stated boost in performance of the proposed semi-supervised method compared to the supervised upper bound model, and although the feasibility and cost-effectiveness are mentioned in the results section, the reviewer suggests a more detailed discussion of the proposed algorithm performance and trade-offs as well as current proposed model challenges in the conclusions section as well.

Reply: We thank the reviewer for his/her valuable work in commenting our paper. We fully agree that the cost-effectiveness of this method is a strong added value for autonomous driving. The model challenges are related to the fact that in real world applications the upperbound model would not be available, hence the improvement can be measured only with respect to the baseline. The availability of well annotated data remains an important issue for deep learning, though semi-supervised learning and co-training clearly help.

Thanks to this reviewer’s comment a more detailed discussion section was added before the Conclusions and a paragraph to better underline this aspect was added to the conclusions.

Action:

A new Subsection 5.5 – Discussion was added to the paper reading:

“The purpose of the experiments proposed in Section 5 is three-fold: (1) to evaluate the potential benefits of using a multi-sensor system over single modality approaches, (2) to investigate the model generalization capabilities to more challenging domains, and (3) to evaluate whether co-training and our newly introduced fusion-based strategy could be useful for domain adaptation.

According to the results shown in Section 5, there are several potential benefits supporting the use of semi-supervised learning, sensor fusion and co-training strategies.

Firstly, a multi-sensor system is shown to provide high reliability and redundancy, for all the cases reported in Table 3 sensor-fusion over performs the single sensor approach (RGB camera or lidar).

Secondly, Table 3 also shows that co-training and semi-supervised learning help the model to generalize knowledge to more challenging domains.

In fact, the best improvement of semi-supervised learning techniques is shown in the night-rain case with over 10% IoU performance increase, starting from 70% of the fusion baseline and reaching 81% in the fusion-semi case.

The use of non-annotated data has clearly shown to improve the overall performance, resulting in big savings in practical applications where data annotation is a heavy and complex task.

However, data annotation cannot be neglected, the upperbound model is shown to be the best performing one, hence full data annotation is still the best way to achieve high reliability and stability in neural networks.”

The following lines were added to the Conclusions:

… “The paper shows that the upperbound model performance level can be reached using other methods such as semi-supervised learning and co-training, resulting in a cost-effective method that uses less data annotation. This result is supported by a cross-validation using 10 different data splits.” …

Round 2

Reviewer 2 Report

Although I don't quite agree with the technical novelty of this paper, the authors have made modifications according to other reviewers' comments.